# Obsessive Compulsive “Paper Handling”: A Potential Distinctive Behavior in Children and Adolescents with KBG Syndrome

**DOI:** 10.3390/jcm11164687

**Published:** 2022-08-11

**Authors:** Francesco Demaria, Paolo Alfieri, Maria Cristina Digilio, Maria Pontillo, Cristina Di Vincenzo, Federica Alice Maria Montanaro, Valentina Ciullo, Giuseppe Zampino, Stefano Vicari

**Affiliations:** 1Child and Adolescent Neuropsychiatry Unit, Department of Neuroscience, Bambino Gesù Children’s Hospital, IRCCS, 00165 Rome, Italy; 2Medical Genetics Unit, Bambino Gesù Children’s Hospital, IRCCS, 00165 Rome, Italy; 3IRCCS Santa Lucia Foundation, Laboratory of Neuropsychiatry, 00179 Rome, Italy; 4Istituto di Clinica Pediatrica, Università Cattolica del Sacro Cuore, 00168 Rome, Italy; 5Department of Life Sciences and Public Health, Università Cattolica del Sacro Cuore, 00168 Rome, Italy

**Keywords:** KBG syndrome, obsessive compulsive disorder/behavior, paper handling

## Abstract

KBG syndrome (KBGS; OMIM #148050) is a rare disease characterized by short stature, facial dysmorphism, macrodontia of the upper central incisors, skeletal anomalies, and neurodevelopmental disorder/intellectual disability. It is caused by a heterozygous variant or 16q24.3 microdeletions of the ANKRD11 gene (OMIM #611192), which plays a primary role in neuronal development. KBGS traits are variable, and mild expressions of the phenotype may complicate diagnosis. The present work aims at improving the characterization of KBGS in order to facilitate its recognition. A psychopathological evaluation of 17 subjects affected by KBGS found that 10 patients exhibited peculiar behavior related to “paper handling”. These children and adolescents performed repetitive activities with paper, reminiscent of the hoarding and ordering behaviors characteristic of obsessive compulsive disorder. Their activities were time consuming and carried out in solitary, and forced interruption could generate intense emotional reactions. Paper handling may thus be understood as a potential distinct KBGS symptom akin to an obsessive compulsive symptom. Further research is needed to verify this claim.

## 1. Introduction

KBG syndrome (KBGS; OMIM #148050) is a rare disease first described in 1975, when it was recognized in seven individuals belonging to three different family groups, but sharing similar features. These individuals presented with short stature, facial dysmorphism, macrodontia of the upper central incisors, skeletal anomalies, and intellectual disability (ID) [1]. The name KBG is comprised of the initials of the surnames of the original three affected families. To date, KBGS has been diagnosed in more than 200 individuals. However, this figure is likely under-representative, as the high variability in KBG features and the possibility that some affected individuals may express a mild phenotype could lead to an underdiagnosis of the disorder [2].

KBG syndrome is caused by a heterozygous variant or 16q24.3 microdeletions of the ANKRD11 gene (OMIM #611192), thereby altering its function. ANKRD11 (ankyrin repeat domain 11) encodes a protein that plays a fundamental role in the co-regulation of the proliferation of neural progenitors in the developing brain, of the genesis and positioning of newborn neurons, of neuronal plasticity, and of dendritic differentiation [3,4,5]. The marked phenotypic variability of KBGS indicates its wide range of expressivity and penetration [6,7].

On a cognitive level, subjects affected by KBGS may display a variety of symptoms, ranging from mild learning difficulties (particularly in females) to moderate ID (predominately in males). The majority of individuals present with mild ID [8,9].

The behavioral phenotype is likewise extremely heterogeneous. Some studies have reported hyperactive behavior (as in attention deficit hyperactivity disorder-ADHD), as well as autistic-like behaviors that, in some subjects, show features of autism spectrum disorder [7,10]. Furthermore, Low et al. [7] described a diverse set of behaviors, including rigidity, fixation on routine, emotional immaturity and reactions of anger and aggression.

Syndromic subjects often behave in characteristic and distinctive ways. For instance, they might insert foreign objects into body orifices (polyembolokoilamania); yank fingernails and/or toenails (onychotillomania, as in Smith-Magenis syndrome) [11]; perform repetitive hand movements (as in Rett syndrome) [12]; make cat-like crying sounds (as in Cri-du-chat syndrome) [13] and practice self-injurious behavior (as in the skin-picking of Prader-Willi syndrome and the face hitting of Cornelia de Lange syndrome) [14]. Identification of these behaviors can support the early recognition of KBGS, especially when symptoms are otherwise variable and mild.

The present work aims to improve the characterization of KBGS expression in order to facilitate the recognition of KBGS phenotypic behavior and allow earlier intervention.

## 2. State of the Art and Our Previous Findings

The psychopathological features of KBGS have been only marginally described in the literature. The majority of works are reports or single-case studies, therefore they do not offer a systematic characterization of the syndrome. Only Van Dongen and colleagues [9] have recently investigated KBGS behavioral symptomatology comparing 18 KBGS patients with 17 other-syndrome patients showing that the former exhibit a greater emotional dysregulation. For this reason, in order to enrich the systematic characterization of KBGS we subsequently carried out a study [15], in which we investigated the psychopathological profile of 17 children, adolescents, and young adults with a molecularly confirmed diagnosis of KBGS. The diagnostic evaluation, which was carried out according to the Diagnostic and Statistical Manual of Mental Disorders 4th edition (DSM-IV-TR) [16], identified several comorbid disorders, including obsessive compulsive disorder (OCD), anxiety spectrum disorders, depressive disorder, ADHD and tic disorder. In our sample, these disorders were present at higher rates than in the general population. Furthermore, OCD and obsessive compulsive (OC) symptoms were the most frequently observed psychopathological conditions (82%; *n* = 14/17). There was no evidence that these data could be correlated to subjects’ cognitive level (measured by the Intelligent Quotient, IQ), which was characterized by a low average IQ (=66), with a variability similar to the one in general population.

OCD is a neuropsychiatric disorder characterized by the presence of obsessive (i.e., intrusive, repetitive, unwanted) thoughts and compulsive behaviors or mental acts [17]. OC symptoms are widespread in the general population. However, when they are present at high levels, they constitute OCD.

OCD has an estimated prevalence of 1–3% in children and adolescents [18], even though the prevalence of OC symptoms in this population is more debated and has been reported in a range of 7–35% [19]. Anxiety and depressive disorders are the most frequent comorbidities of OCD and OC symptoms [17]. OC symptoms can manifest as obsessive concern over dirt/contamination by pathogenic germs, aggressive behavior (i.e., fear of harming oneself, fear of harming others), and sexual/religious ideas. The most frequent compulsions include environmental cleaning/hand washing, checking, ordering, repetition, mental compulsions (e.g., counting), and hoarding [20,21]. Obsessions and compulsions can be time-consuming (often involving more than 1 h per day) and they may significantly interfere with normal life habits, thereby compromising the quality of life [20,22].

In our research [15], we found that our KBGS patients exhibited a higher prevalence of OCD in comparison to the age-matched unaffected population, with compulsion being more widespread than obsessions.

Hence, we subsequently decided to further analyze the obtained data in order to better investigate if some obsessions/compulsions were more frequent than others. More specifically, we observed a prevalence of hoarding (*n* = 11/17), ordering and checking (both *n* = 6/17) OCD behavioral symptoms. In particular, a peculiar behavior was noted in relation to “paper handling,” which was both observed and confirmed by the subjects’ parents. These activities were exhibited in various ways and included cutting, handling, folding, straining, crumpling, and hoarding pieces of paper. This action was neither playful nor pleasant, but repetitive and poorly finalized. As many as 10 out of 17 patients (58.8%) used to perform these paper-related activities and one patient collected/hoarded stickers. In patients who displayed paper handling, the paper handling was associated with OCD behavioral symptoms: eight presented with a hoarding symptom, (of these) six presented hoarding and ordering symptoms, while two presented with a checking symptom (Table 1). Consequently, we speculated that paper handling activities could be a possible distinctive behavior in KBG syndrome. Therefore, we decided to continue enrolling KBG patients, with the aim of generating a better characterization of their OCD behavioral symptoms.

## 3. The Present Study

Considering our previous finding about the widespread presence of OC symptoms in KBGS [15], we carried out further evaluations of other four children and adolescents diagnosed with KBGS. Participants were assessed by the same team (comprised of an expert child psychiatrist and experienced child psychologists) as used in the first study. These clinicians investigated psychopathological signs/psychiatric disorders in participants, according to the criteria of the Diagnostic and Statistical Manual of Mental Disorders, fifth edition.

### 3.1. Methods

In addition to the neuropsychological assessment, which has been performed in order to have complete profiling of our KBGS sample, we carried out a systematic psychopathological evaluation with the aim of further analyzing the psychological and behavioral features in KBGS patients. Cognitive functioning was measured using appropriate developmental instruments, namely Wechsler Intelligence Scale for Children Fourth Edition (WISC-IV) [23], Wechsler Adult Intelligence Scale Fourth Edition (WAIS-IV) [24] and Leiter International Performance Scale Third edition [25].

Psychopathological features were investigated with the following standardized tools:Schedule for Affective Disorders and Schizophrenia for School Age Children, Present and Lifetime version (KSADS PL), a semi-structured interview for children and their parents that allows for the detection of current and lifetime psychopathological/psychiatric symptoms according to DSM-5 criteria [26].Children’s Yale–Brown Obsessive Compulsive Scale (CY–BOCS) [27], which is a self/parent report questionnaire designed to rate the typology and the severity of obsessive and compulsive symptoms.

The selection of the above tools was endorsed by the acknowledgment that the American Academy of Child & Adolescent Psychiatry (AACAP) defines these instruments as a gold standard for the evaluation of psychiatric symptoms in developmental age, including them in its guidelines and Practice Parameters [28].

### 3.2. Results

As in our previous research [15], we observed a high prevalence of repetitive activities, which were again confirmed by parents; in particular, three out of four subjects with a confirmed molecular diagnosis of KBGS performed paper handling (75%), including tearing off (one subject), crumpling and hoarding the pieces of paper. Again, paper handling was present in comorbidity with OCD behavioral symptoms, such as hoarding and hoarding and ordering. *(*Table 2).

Parents described paper handling as a time-consuming activity that was generally carried out alone. They also indicated that interruptions in this activity could generate intense emotional reactions.

Paper handling may be comparable to an OC symptom, according to the diagnostic criteria of the DSM-IV-TR and DSM-5 [16,17]. In particular, the association with hoarding, ordering, and checking would support this association. According to our clinical evaluations, the paper handling behavior did not resemble stereotypy (as generally observed in autistic patients), as the assessed subjects had linguistic appropriateness and uncompromised communication and social interaction. Furthermore, even though their cognitive level was averagely low, again we did not find correlations between IQ and OC symptoms.

## 4. Conclusions and Future Work

Our further evaluations suggest that “paper handling” may represent a potential distinctive behavior of KBG syndrome. Mutations of the ANKRD11 gene (which is associated with KBGS) have also been identified in individuals with Cornelia de Lange syndrome and other developmental disorders, with this phenotypic overlap increasing the challenge of clinical diagnosis [29,30]. For this reason, the characterization of the cognitive-behavioral phenotype in genetic syndromes is essential, as it may help to find out a specific correlation between the genetic background and psychological and behavioral features. Indeed, the identification of the specific behavioral characteristics of KBGS could facilitate better recognition of the syndrome, help to distinguish it from other genetic syndromes, improve treatment and enhance the delivery of comprehensive support for families. The present profiling provides important information that clinicians could take into account during their clinical activity. On the other hand, some important limitations should be considered in our study. The characteristic behavior “paper handling” in KBG syndrome needs to be confirmed by further work using a control group. Indeed, a comparison with a control group (for instance a group of individuals with Prader-Willi syndrome who exhibit ritualistic behaviors as well [31] or a group of patients with OCD) and a greater understanding of repetitive and stereotyped behaviors are necessary to better clarify the nature of “paper handling” and confirm this evidence. Additionally, the study of the behavior and psychopathology of rare and syndromic diseases is a new frontier to be discovered. Future studies on larger cohorts and longitudinal analysis on the subject become necessary.

## Figures and Tables

**Table 1 jcm-11-04687-t001:** Obsessive compulsive symptoms and “paper handling” activities in 17 KBGS (KBG syndrome) subjects; OC = obsessive-compulsive.

OC Symptoms	KBG (*N* = 17) of Which	KBG Paper Handling (*n* = 10)
Hoarding	11	8
Ordering	6	-
Hoarding & ordering	-	6
Checking	6	2
Washing & cleaning	3	-
Contamination	1	-

**Table 2 jcm-11-04687-t002:** Obsessive compulsive symptoms and “paper handling” activities in 4 newly diagnosed KBGS (KBG syndrome) subjects. OC = obsessive-compulsive).

OC Symptoms	KBG (*N* = 4) of Which	KBG Paper Handling (*n* = 3)
Hoarding	3	2
Ordering	-	-
Hoarding & ordering	-	1

## Data Availability

The raw data supporting the conclusions of this article will be made available by the authors, without undue reservation.

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
