# Peer review of "Obsessive Compulsive “Paper Handling”: A Potential Distinctive Behavior in Children and Adolescents with KBG Syndrome"

_jcm, 2022, doi:10.3390/jcm11164687_

Round 1

Reviewer 1 Report

There are some serious flaws in the manuscript. I am aware that KBG is very rare , but only four participants were studied (negligible statistical power, no generalization possible etc.). In addition, lack of an adequate control group (e.g., people with OCD and healthy controls). This dilutes the conclusions drawn.

Reviewer 2 Report

The authors identified a higher rate of obsessive-compulsive symptoms, especially a peculiar behavior which they called as paper handling behavior in patients with a molecularly confirmed diagnosis of KBG syndrome. In the present study, they did an in-depth exploration of psychopathological and OCD features in a subset of those patients (n=4). They could find that three out of four of them had paper handling behavior which can be taken as a distinctive clinical phenotype of the KBG syndrome. The study method is adequate for the purpose and the conclusion is valid based on the results obtained. The authors have identified possible areas where the current findings would be helpful.

Reviewer 3 Report

The manuscript reports a specific evaluation of a group of patients with KGB syndrome. The authors have deeper evaluated a group of patients already included in a study, reporting a specific pattern of paper handling as specifically related to this syndrome. The authors have the merit to work on a rare disorder, looking for information that could be helpful in the clinical management of the patients. I think it needs some changes. 

- the end of the "2. state of art" is not so clear. The authors seem to anticipate the results of the paper. I think they should revise this part. 

- it is not clear to me if the evaluation of the patients was successive than the first one, and in that case after how many times? If it was part of the data collected for the first study, my question would be: why did you create a salami paper? 

- methodology is very concise.

- was the evaluations carried out by the same people? 

Round 2

Reviewer 1 Report

My comments have not been adequatley adressed. In their response the authors state that 'the study aimed at providing information on a behavior that is unique to this population'. However, such conclusions can't be made without an adequate control group (which is still missing).

Author Response

We really thank the reviewer for his observation. We modified the conclusions in the paper according to the received suggestions.

Reviewer 3 Report

I think the manuscript has been revised. 

Author Response

We really thank you for your comments and reply.